# Working hard to know your neighbor's margins: Local descriptor learning loss

**Anastasiya Mishchuk[1], Dmytro Mishkin[2], Filip Radenović[2], Jiři Matas[2]**

[1] Szkocka Research Group, Ukraine
anastasiya.mishchuk@gmail.com

[2] Visual Recognition Group, CTU in Prague
{mishkdmy, filip.radenovic, matas}@cmp.felk.cvut.cz

## Abstract

We introduce a loss for metric learning, which is inspired by the Lowe's matching criterion for SIFT. We show that the proposed loss, that maximizes the distance between the closest positive and closest negative example in the batch, is better than complex regularization methods; it works well for both shallow and deep convolution network architectures. Applying the novel loss to the L2Net CNN architecture results in a compact descriptor named HardNet. It has the same dimensionality as SIFT (128) and shows state-of-art performance in wide baseline stereo, patch verification and instance retrieval benchmarks.

## 1 Introduction

Many computer vision tasks rely on finding local correspondences, e.g. image retrieval [1, 2], panorama stitching [3], wide baseline stereo [4], 3D-reconstruction [5, 6]. Despite the growing number of attempts to replace complex classical pipelines with end-to-end learned models, *e.g.*, for image matching [7], camera localization [8], the classical detectors and descriptors of local patches are still in use, due to their robustness, efficiency and their tight integration. Moreover, reformulating the task, which is solved by the complex pipeline as a differentiable end-to-end process is highly challenging.

As a first step towards end-to-end learning, hand-crafted descriptors like SIFT [9, 10] or detectors [9, 11, 12] have been replace with learned ones, e.g., LIFT [13], MatchNet [14] and DeepCompare [15]. However, these descriptors have not gained popularity in practical applications despite good performance in the patch verification task. Recent studies have confirmed that SIFT and its variants (RootSIFT-PCA [16], DSP-SIFT [17]) significantly outperform learned descriptors in image matching and small-scale retrieval [18], as well as in 3D-reconstruction [19]. One of the conclusions made in [19] is that current local patches datasets are not large and diverse enough to allow the learning of a high-quality widely-applicable descriptor.

In this paper, we focus on descriptor learning and, using a novel method, train a convolutional neural network (CNN), called HardNet. We additionally show that our learned descriptor significantly outperforms both hand-crafted and learned descriptors in real-world tasks like image retrieval and two view matching under extreme conditions. For the training, we use the standard patch correspondence data thus showing that the available datasets are sufficient for going beyond the state of the art.

## 2 Related work

Classical SIFT local feature matching consists of two parts: finding nearest neighbors and comparing the first to second nearest neighbor distance ratio threshold for filtering false positive matches. To

best of our knowledge, no work in local descriptor learning fully mimics such strategy as the learning objective.

Simonyan and Zisserman [20] proposed a simple filter plus pooling scheme learned with convex optimization to replace the hand-crafted filters and poolings in SIFT. Han et al. [14] proposed a two-stage siamese architecture – for embedding and for two-patch similarity. The latter network improved matching performance, but prevented the use of fast approximate nearest neighbor algorithms like kd-tree [21]. Zagoruyko and Komodakis [15] have independently presented similar siamese-based method which explored different convolutional architectures. Simo-Serra et al [22] harnessed hard-negative mining with a relative shallow architecture that exploited pair-based similarity.

The three following papers have most closely followed the classical SIFT matching scheme. Balntas et al [23] used a triplet margin loss and a triplet distance loss, with random sampling of the patch triplets. They show the superiority of the triplet-based architecture over a pair based. Although, unlike SIFT matching or our work, they sampled negatives randomly. Choy et al [7] calculate the distance matrix for mining positive as well as negative examples, followed by pairwise contrastive loss.

Tian et al [24] use $n$ matching pairs in batch for generating $n^2 - n$ negative samples and require that the distance to the ground truth matchings is minimum in each row and column. No other constraint on the distance or distance ratio is enforced. Instead, they propose a penalty for the correlation of the descriptor dimensions and adopt deep supervision [25] by using intermediate feature maps for matching. Given the state-of-art performance, we have adopted the L2Net [24] architecture as base for our descriptor. We show that it is possible to learn even more powerful descriptor with significantly simpler learning objective without need of the two auxiliary loss terms.

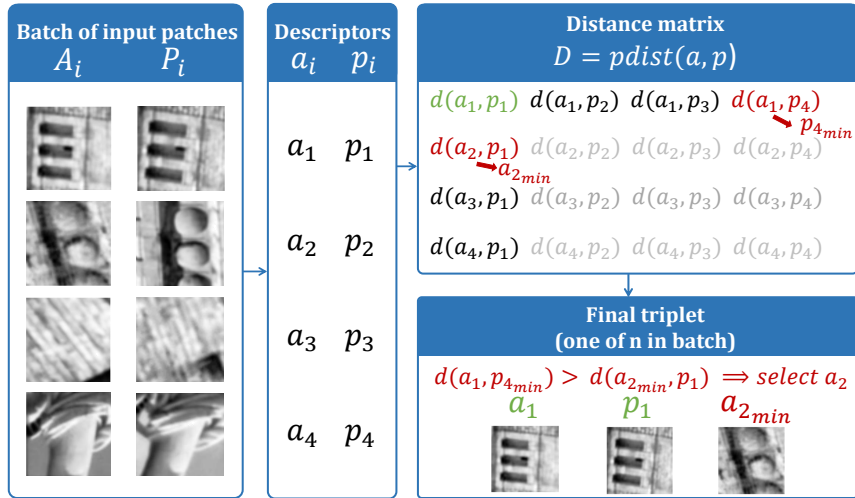

Figure 1: Proposed sampling procedure. First, patches are described by the current network, then a distance matrix is calculated. The closest non-matching descriptor – shown in red – is selected for each $a_i$ and $p_i$ patch from positive pair (green) respectively. Finally, among two negative candidates the hardest one is chosen. All operations are done in a single forward pass.

## 3 The proposed descriptor

### 3.1 Sampling and loss

Our learning objective mimics SIFT matching criterion. The process is shown in Figure 1. First, a batch $\mathcal{X} = (A_i, P_i)_{i=1..n}$ of matching local patches is generated, where $A$ stands for the anchor and $P$ for the positive. The patches $A_i$ and $P_i$ correspond to the same point on 3D surface. We make sure that in batch $\mathcal{X}$, there is exactly one pair originating from a given 3D point.

Second, the $2n$ patches in $\mathcal{X}$ are passed through the network shown in Figure 2.

L2 pairwise distance matrix $D = \mathrm{cdist}(a, p)$, where, $d(a_i, p_j) = \sqrt{2 - 2a_i p_j}$, $i = 1..n, j = 1..n$ of size $n \times n$ is calculated, where $a_i$ and $p_j$ denote the descriptors of patches $A_i$ and $P_j$ respectively.

Next, for each matching pair $a_i$ and $p_i$ the closest non-matching descriptors i.e. the $2^{nd}$ nearest neighbor, are found respectively:

$a_i$ – anchor descriptor, $p_i$ – positive descriptor,

$p_{j_{min}}$ – closest non-matching descriptor to $a_i$, where $j_{min} = \arg\min_{j=1..n, j \neq i} d(a_i, p_j)$,

$a_{k_{min}}$ – closest non-matching descriptor to $p_i$ where $k_{min} = \arg\min_{k=1..n, k \neq i} d(a_k, p_i)$.

Then from each quadruplet of descriptors $(a_i, p_i, p_{j_{min}}, a_{k_{min}})$, a triplet is formed: $(a_i, p_i, p_{j_{min}})$, if $d(a_i, p_{j_{min}}) < d(a_{k_{min}}, p_i)$ and $(p_i, a_i, a_{k_{min}})$ otherwise.

Our goal is to minimize the distance between the matching descriptor and closest non-matching descriptor. These $n$ triplet distances are fed into the triplet margin loss:

$$L = \frac{1}{n} \sum_{i=1,n} \max\left(0, 1 + d(a_i, p_i) - \min\left(d(a_i, p_{j_{min}}), d(a_{k_{min}}, p_i)\right)\right) \quad (1)$$

where $\min\left(d(a_i, p_{j_{min}}), d(a_{k_{min}}, p_i)\right)$ is pre-computed during the triplet construction.

The distance matrix calculation is done on GPU and the only overhead compared to the random triplet sampling is the distance matrix calculation and calculating the minimum over rows and columns. Moreover, compared to usual learning with triplets, our scheme needs only two-stream CNN, not three, which results in 30% less memory consumption and computations.

Unlike in [24], neither deep supervision for intermediate layers is used, nor a constraint on the correlation of descriptor dimensions. We experienced no significant over-fitting.

## 3.2 Model architecture

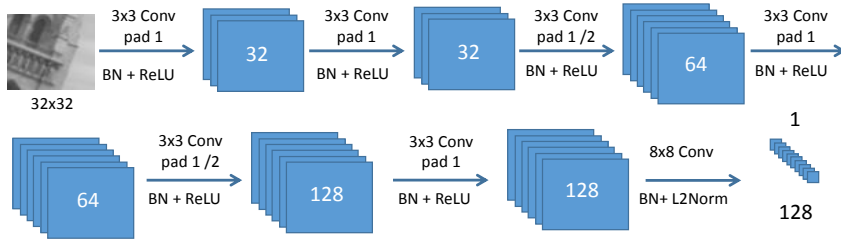

Figure 2: The architecture of our network, adopted from L2Net [24]. Each convolutional layer is followed by batch normalization and ReLU, except the last one. Dropout regularization is used before the last convolution layer.

The HardNet architecture, Figure 2, is identical to L2Net [24]. Padding with zeros is applied to all convolutional layers, to preserve the spatial size, except to the final one. There are no pooling layers, since we found that they decrease performance of the descriptor. That is why the spatial size is reduced by strided convolutions. Batch normalization [26] layer followed by ReLU [27] non-linearity is added after each layer, except the last one. Dropout [28] regularization with 0.1 dropout rate is applied before the last convolution layer. The output of the network is L2 normalized to produce 128-D descriptor with unit-length. Grayscale input patches with size $32 \times 32$ pixels are normalized by subtracting the per-patch mean and dividing by the per-patch standard deviation.

Optimization is done by stochastic gradient descent with learning rate of 0.1, momentum of 0.9 and weight decay of 0.0001. Learning rate was linearly decayed to zero within 10 epochs for the most of the experiments in this paper. Training is done with PyTorch library [29].

## 3.3 Model training

UBC Phototour [3], also known as Brown dataset. It consists of three subsets: *Liberty*, *Notre Dame* and *Yosemite* with about 400k normalized 64x64 patches in each. Keypoints were detected by DoG detector and verified by 3D model.

Test set consists of 100k matching and non-matching pairs for each sequence. Common setup is to train descriptor on one subset and test on two others. Metric is the false positive rate (FPR) at point of 0.95 true positive recall. It was found out by Michel Keller that [14] and [23] evaluation procedure reports FDR (false discovery rate) instead of FPR (false positive rate). To avoid the incomprehension of results we've decided to provide both FPR and FDR rates and re-estimated the scores for straight comparison. Results are shown in Table 1. Proposed descriptor outperforms competitors, with training augmentation, or without it. We haven't included results on multiscale patch sampling or so called "center-surrounding" architecture for two reasons. First, architectural choices are beyond the scope of current paper. Second, it was already shown in [24, 30] that "center-surrounding" consistently improves results on Brown dataset for different descriptors, while hurts matching performance on other, more realistic setups, *e.g.*, on Oxford-Affine [31] dataset.

In the rest of paper we use descriptor trained on *Liberty* sequence, which is a common practice, to allow a fair comparison. TFeat [23] and L2Net [24] use the same dataset for training.

Table 1: Patch correspondence verification performance on the Brown dataset. We report false positive rate at true positive rate equal to 95% (FPR95). **Some papers report false discovery rate (FDR) instead of FPR due to bug in the source code**. For consistency we provide FPR, either obtained from the original article or re-estimated from the given FDR (marked with *). The best results are in **bold**.

| Training | Notredame | Yosemite | Liberty | Yosemite | Liberty | Notredame | Mean | |
|---|---|---|---|---|---|---|---|---|
| Test | Liberty | | Notredame | | Yosemite | | FDR | FPR |
| SIFT [9] | 29.84 | | 22.53 | | 27.29 | | | 26.55 |
| MatchNet*[14] | 7.04 | 11.47 | 3.82 | 5.65 | 11.6 | 8.7 | 7.74 | 8.05 |
| TFeat-M* [23] | 7.39 | 10.31 | 3.06 | 3.8 | 8.06 | 7.24 | 6.47 | 6.64 |
| L2Net [24] | 3.64 | 5.29 | 1.15 | 1.62 | 4.43 | 3.30 | | 3.24 |
| HardNet (ours) | **3.06** | **4.27** | **0.96** | **1.4** | **3.04** | **2.53** | **3.00** | **2.54** |
| Augmentation: flip, 90° random rotation | | | | | | | | |
| GLoss+[30] | 3.69 | 4.91 | 0.77 | 1.14 | 3.09 | 2.67 | | 2.71 |
| DC2ch2st+[15] | 4.85 | 7.2 | 1.9 | 2.11 | 5.00 | 4.10 | | 4.19 |
| L2Net+ [24] + | 2.36 | 4.7 | 0.72 | 1.29 | 2.57 | **1.71** | | 2.23 |
| HardNet+ (ours) | **2.28** | **3.25** | **0.57** | **0.96** | **2.13** | 2.22 | **1.97** | **1.9** |

## 3.4 Exploring the batch size influence

We study the influence of mini-batch size on the final descriptor performance. It is known that small mini-batches are beneficial to faster convergence and better generalization [32], while large batches allow better GPU utilization. Our loss function design should benefit from seeing more hard negative patches to learn to distinguish them from true positive patches. We report the results for batch sizes 16, 64, 128, 512, 1024, 2048. We trained the model described in Section 3.2 using *Liberty* sequence of Brown dataset. Results are shown in Figure 3. As expected, model performance improves with increasing the mini-batch size, as more examples are seen to get harder negatives. Although, increasing batch size to more than 512 does not bring significant benefit.

## 4 Empirical evaluation

Recently, Balntas et al. [23] showed that good performance on patch verification task on Brown dataset does not always mean good performance in the nearest neighbor setup and vice versa. Therefore, we have extensively evaluated learned descriptors on real-world tasks like two view matching and image retrieval.

We have selected RootSIFT [10], TFeat-M* [23], and L2Net [24] for direct comparison with our descriptor, as they show the best results on a variety of datasets.

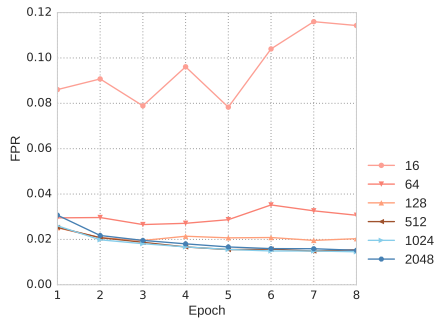

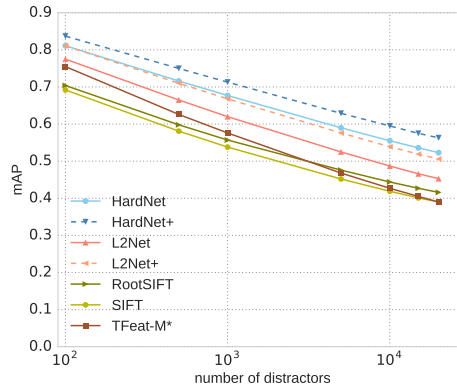

Figure 3: Influence of the batch size on descriptor performance. The metric is false positive rate (FPR) at true positive rate equal to 95%, averaged over *Notredame* and *Yosemite* validation sequences.

Figure 4: Patch retrieval descriptor performance (mAP) vs. the number of distractors, evaluated on HPatches dataset.

## 4.1 Patch descriptor evaluation

HPatches [18] is a recent dataset for local patch descriptor evaluation. It consists of 116 sequences of 6 images. The dataset is split into two parts: *viewpoint* – 59 sequences with significant viewpoint change and *illumination* – 57 sequences with significant illumination change, both natural and artificial. Keypoints are detected by DoG, Hessian and Harris detectors in the reference image and reprojected to the rest of the images in each sequence with 3 levels of geometric noise: *Easy*, *Hard*, and *Tough* variants. The HPatches benchmark defines three tasks: patch correspondence verification, image matching and small-scale patch retrieval. We refer the reader to the HPatches paper [18] for a detailed protocol for each task.

Results are shown in Figure 5. L2Net and HardNet have shown similar performance on the patch verification task with a small advantage of HardNet. On the matching task, even the non-augmented version of HardNet outperforms the augmented version of L2Net+ by a noticeable margin. The difference is larger in the TOUGH and HARD setups. Illumination sequences are more challenging than the geometric ones, for all the descriptors. We have trained network with TFeat architecture, but with proposed loss function – it is denoted as HardTFeat. It outperforms original version in matching and retrieval, while being on par with it on patch verification task.

In patch retrieval, relative performance of the descriptors is similar to the matching problem: HardNet beats L2Net+. Both descriptors significantly outperform the previous state-of-the-art, showing the superiority of the selected deep CNN architecture over the shallow TFeat model.

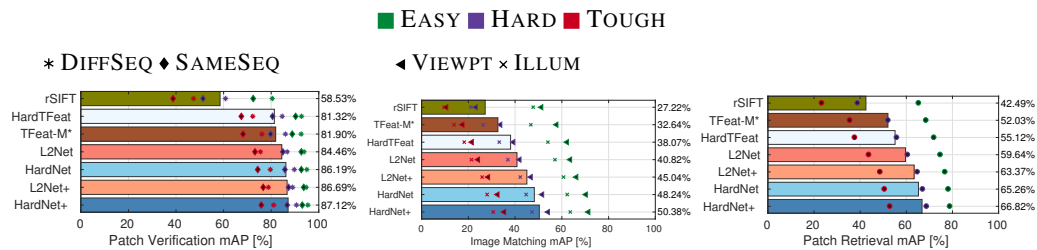

Figure 5: Left to right: Verification, matching and retrieval results on HPatches dataset. Marker color indicates the level of geometrical noise in: EASY, HARD and TOUGH. Marker type indicates the experimental setup. DIFFSEQ and SAMESEQ shows the source of negative examples for the verification task. VIEWPT and ILLUM indicate the type of sequences for matching. None of the descriptors is trained on HPatches.

Table 2: Comparison of the loss functions and sampling strategies on the HPatches matching task, the mean mAP is reported. CPR stands for the regularization penalty of the correlation between descriptor channels, as proposed in [24]. Hard negative mining is performed once per epoch. Best results are in **bold**. HardNet uses the hardest-in-batch sampling and the triplet margin loss.

| Sampling / Loss | Softmin | Triplet margin | Contrastive | |
|---|---|---|---|---|
| | | $m = 1$ | $m = 1$ | $m = 2$ |
| Random | | overfit | | |
| Hard negative mining | | overfit | | |
| Random + CPR | 0.349 | 0.286 | 0.007 | 0.083 |
| Hard negative mining + CPR | 0.391 | 0.346 | 0.055 | 0.279 |
| Hardest in batch (ours) | 0.474 | **0.482** | 0.444 | **0.482** |

We also ran another patch retrieval experiment, varying the number of distractors (non-matching patches) in the retrieval dataset. The results are shown in Figure 4. TFeat descriptor performance, which is comparable to L2Net in the presence of low number distractors, degrades quickly as the size of the the database grows. At about 10,000 its performance drops below SIFT. This experiment explains why TFeat performs relatively poorly on the Oxford5k [33] and Paris6k [34] benchmarks, which contain around 12M and 15M distractors, respectively, see Section 4.4 for more details. Performance of the HardNet decreases slightly for both augmented and plain version and the difference in mAP to other descriptors grows with the increasing complexity of the task.

## 4.2  Ablation study

For better understanding of the significance of the sampling strategy and the loss function, we conduct experiments summarized in Table 2. We train our HardNet model (architecture is exactly the same as L2Net model), change one parameter at a time and evaluate its impact.

The following sampling strategies are compared: random, the proposed "hardest-in-batch", and the "classical" hard negative mining, i.e. selecting in each epoch the closest negatives from the full training set. The following loss functions are tested: softmin on distances, triplet margin with margin $m = 1$, contrastive with margins $m = 1, m = 2$. The last is the maximum possible distance for unit-normed descriptors. Mean mAP on HPatches Matching task is shown in Table 2.

The proposed "hardest-in-batch" clearly outperforms all the other sampling strategies for all loss functions and it is the main reason for HardNet's good performance. The random sampling and "classical" hard negative mining led to huge overfit, when training loss was high, but test performance was low and varied several times from run to run. This behavior was observed with all loss function. Similar results for random sampling were reported in [24]. The poor results of hard negative mining ("hardest-in-the-training-set") are surprising. We guess that this is due to dataset label noise, the mined "hard negatives" are actually positives. Visual inspection confirms this. We were able to get

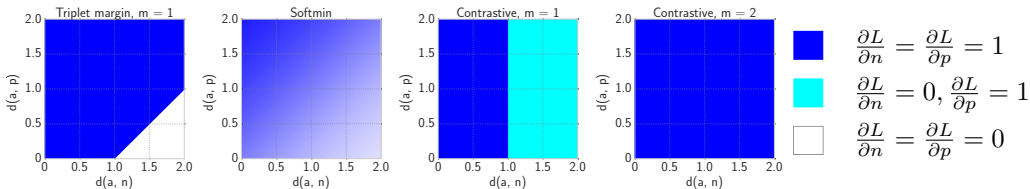

Figure 6: Contribution to the gradient magnitude from the positive and negative examples. Horizontal and vertical axes show the distance from the anchor (a) to the negative (n) and positive (p) examples respectively. Softmin loss gradient quickly decreases when $d(a, n) > d(a, p)$, unlike the triplet margin loss. For the contrastive loss, negative examples with $d(a, n) > m$ contribute zero to the gradient. The triplet margin loss and the contrastive loss with a big margin behave very similarly.

reasonable results with random and hard negative mining sampling only with additional correlation penalty on descriptor channels (CPR), as proposed in [24].

Regarding the loss functions, softmin gave the most stable results across all sampling strategies, but it is marginally outperformed by contrastive and triplet margin loss for our strategy. One possible explanation is that the triplet margin loss and contrastive loss with a large margin have constant non-zero derivative w.r.t to both positive and negative samples, see Figure 6. In the case of contrastive loss with a small margin, many negative examples are not used in the optimization (zero derivatives), while the softmin derivatives become small, once the distance to the positive example is smaller than to the negative one.

### 4.3 Wide baseline stereo

To validate descriptor generalization and their ability to operate in extreme conditions, we tested them on the W1BS dataset [4]. It consists of 40 image pairs with one particular extreme change between the images:

**Appearance (A):** difference in appearance due to seasonal or weather change, occlusions, etc;

**Geometry (G):** difference in scale, camera and object position;

**Illumination (L):** significant difference in intensity, wavelength of light source;

**Sensor (S):** difference in sensor data (IR, MRI).

Moreover, local features in W1BS dataset are detected with MSER [35], Hessian-Affine [11] (in implementation from [36]) and FOCI [37] detectors. They fire on different local structures than DoG. Note that DoG patches were used for the training of the descriptors. Another significant difference to the HPatches setup is the absence of the geometrical noise: all patches are perfectly reprojected to the target image in pair. The testing protocol is the same as for the HPatches matching task.

Results are shown in Figure 7. HardNet and L2Net perform comparably, former is performing better on images with geometrical and appearance changes, while latter works a bit better in map2photo and visible-vs-infrared pairs. Both outperform SIFT, but only by a small margin. However, considering the significant amount of the domain shift, descriptors perform very well, while TFeat loses badly to SIFT. HardTFeat significantly outperforms the original TFeat descriptor on the W1BS dataset, showing the superiority of the proposed loss.

Good performance on patch matching and verification task does not automatically lead to the better performance in practice, e.g. to more images registered. Therefore we also compared descriptor on wide baseline stereo setup with two metric: number of successfully matched image pairs and average number of inliers per matched pair, following the matcher comparison protocol from [4]. The only change to the original protocol is that first fast matching steps with ORB detector and descriptor were removed, as we are comparing "SIFT-replacement" descriptors.

The results are shown in Table 3. Results on Edge Foci (EF) [37], Extreme view [38] and Oxford Affine [11] datasets are saturated and all the descriptors are good enough for matching all image pairs.

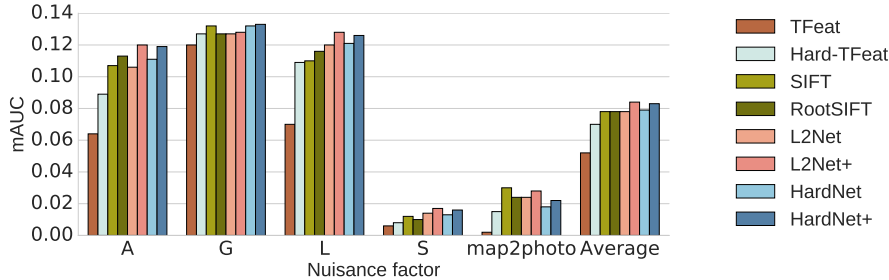

Figure 7: Descriptor evaluation on the W1BS patch dataset, mean area under precision-recall curve is reported. Letters denote nuisance factor, A: appearance; G: viewpoint/geometry; L: illumination; S: sensor; map2photo: satellite photo vs. map.

HardNet has an a slight advantage in a number of inliers per image. The rest of datasets: SymB [39], GDB [40], WxBS [4] and LTLL [41] have one thing in common: image pairs are or from different domain than photo (e.g. drawing to drawing) or cross-domain (e.g., drawing to photo). Here HardNet outperforms learned descriptors and is on-par with hand-crafted RootSIFT. We would like to note that HardNet was not learned to match in different domain, nor cross-domain scenario, therefore such results show the generalization ability.

Table 3: Comparison of the descriptors on wide baseline stereo within MODS matcher[4] on wide baseline stereo datasets. Number of matched image pairs and average number of inliers are reported. Numbers is the header corresponds to the number of image pairs in dataset.

| | EF | | EVD | | OxAff | | SymB | | GDB | | WxBS | | LTLL | |
|---|---|---|---|---|---|---|---|---|---|---|---|---|---|---|
| Descriptor | 33 | inl. | 15 | inl. | 40 | inl. | 46 | inl. | 22 | inl. | 37 | inl. | 172 | inl. |
| RootSIFT | **33** | 32 | **15** | 34 | **40** | 169 | **45** | 43 | **21** | 52 | **11** | **93** | 123 | 27 |
| TFeat-M* | 32 | 30 | **15** | 37 | **40** | 265 | 40 | 45 | 16 | 72 | 10 | 62 | 96 | 29 |
| L2Net+ | **33** | 34 | **15** | 34 | **40** | 304 | 43 | 46 | 19 | **78** | 9 | 51 | **127** | 26 |
| HardNet+ | **33** | 35 | **15** | 41 | **40** | 316 | 44 | **47** | **21** | 75 | **11** | 54 | **127** | **31** |

## 4.4 Image retrieval

We evaluate our method, and compare against the related ones, on the practical application of image retrieval with local features. Standard image retrieval datasets are used for the evaluation, *i.e.*, Oxford5k [33] and Paris6k [34] datasets. Both datasets contain a set of images (5062 for Oxford5k and 6300 for Paris6k) depicting 11 different landmarks together with distractors. For each of the 11 landmarks there are 5 different query regions defined by a bounding box, constituting 55 query regions per dataset. The performance is reported as mean average precision (mAP) [33].

In the first experiment, for each image in the dataset, multi-scale Hessian-affine features [31] are extracted. Exactly the same features are described by ours and all related methods, each of them producing a 128-D descriptor per feature. Then, k-means with approximate nearest neighbor [21] is used to learn a 1 million visual vocabulary on an independent dataset, that is, when evaluating on Oxford5k, the vocabulary is learned with descriptors of Paris6k and *vice versa*. All descriptors of testing dataset are assigned to the corresponding vocabulary, so finally, an image is represented by the histogram of visual word occurrences, *i.e.*, the bag-of-words (BoW) [1] representation, and an inverted file is used for an efficient search. Additionally, spatial verification (SV) [33], and standard query expansion (QE) [34] are used to re-rank and refine the search results. Comparison with the related work on patch description is presented in Table 4. HardNet+ and L2Net+ perform comparably across both datasets and all settings, with slightly better performance of HardNet+ on average across

Table 4: Performance (mAP) evaluation on bag-of-words (BoW) image retrieval. Vocabulary consisting of 1M visual words is learned on independent dataset, that is, when evaluating on Oxford5k, the vocabulary is learned with features of Paris6k and *vice versa*. SV: spatial verification. QE: query expansion. The best results are highlighted in **bold**. All the descriptors except SIFT and HardNet++ were learned on *Liberty* sequence of Brown dataset [3]. HardNet++ is trained on union of Brown and HPatches [18] datasets.

| | Oxford5k | | | Paris6k | | |
|---|---|---|---|---|---|---|
| Descriptor | BoW | BoW+SV | BoW+QE | BoW | BoW+SV | BoW+QE |
| TFeat-M* [23] | 46.7 | 55.6 | 72.2 | 43.8 | 51.8 | 65.3 |
| RootSIFT [10] | 55.1 | 63.0 | 78.4 | 59.3 | 63.7 | 76.4 |
| L2Net+ [24] | **59.8** | 67.7 | 80.4 | **63.0** | 66.6 | 77.2 |
| HardNet | 59.0 | 67.6 | **83.2** | 61.4 | **67.4** | **77.5** |
| HardNet+ | **59.8** | 68.8 | 83.0 | 61.0 | 67.0 | **77.5** |
| HardNet++ | **60.8** | **69.6** | **84.5** | **65.0** | **70.3** | **79.1** |

Table 5: Performance (mAP) comparison with the state-of-the-art image retrieval with local features. Vocabulary is learned on independent dataset, that is, when evaluating on Oxford5k, the vocabulary is learned with features of Paris6k and *vice versa*. All presented results are with spatial verification and query expansion. VS: vocabulary size. SA: single assignment. MA: multiple assignments. The best results are highlighted in **bold**.

| Method | VS | Oxford5k | | Paris6k | |
|---|---|---|---|---|---|
| | | SA | MA | SA | MA |
| SIFT–BoW [36] | 1M | 78.4 | 82.2 | – | – |
| SIFT–BoW-fVocab [46] | 16M | 74.0 | 84.9 | 73.6 | 82.4 |
| RootSIFT–HQE [43] | 65k | 85.3 | 88.0 | 81.3 | 82.8 |
| HardNet++–HQE | 65k | **86.8** | **88.3** | **82.8** | **84.9** |

all results (average mAP 69.5 vs. 69.1). RootSIFT, which was the best performing descriptor in image retrieval for a long time, falls behind with average mAP 66.0 across all results.

We also trained HardNet++ version – with all available training data at the moment: union of Brown and HPatches datasets, instead of just *Liberty* sequence from Brown for the HardNet+. It shows the benefits of having more training data and is performing best for all setups.

Finally, we compare our descriptor with the state-of-the-art image retrieval approaches that use local features. For fairness, all methods presented in Table 5 use the same local feature detector as described before, learn the vocabulary on an independent dataset, and use spatial verification (SV) and query expansion (QE). In our case (HardNet++–HQE), a visual vocabulary of 65k visual words is learned, with additional Hamming embedding (HE) [42] technique that further refines descriptor assignments with a 128 bits binary signature. We follow the same procedure as RootSIFT–HQE [43] method, by replacing RootSIFT with our learned HardNet++ descriptor. Specifically, we use: (i) weighting of the votes as a decreasing function of the Hamming distance [44]; (ii) burstiness suppression [44]; (iii) multiple assignments of features to visual words [34, 45]; and (iv) QE with feature aggregation [43]. All parameters are set as in [43]. The performance of our method is the best reported on both Oxford5k and Paris6k when learning the vocabulary on an independent dataset (mAP 89.1 was reported [10] on Oxford5k by learning it on the same dataset comprising the relevant images), and using the same amount of features (mAP 89.4 was reported [43] on Oxford5k when using twice as many local features, *i.e.*, 22M compared to 12.5M used here).

## 5    Conclusions

We proposed a novel loss function for learning a local image descriptor that relies on the hard negative mining within a mini-batch and the maximization of the distance between the closest positive and closest negative patches. The proposed sampling strategy outperforms classical hard-negative mining and random sampling for softmin, triplet margin and contrastive losses.

The resulting descriptor is compact – it has the same dimensionality as SIFT (128), it shows state-of-art performance on standard matching, patch verification and retrieval benchmarks and it is fast to compute on a GPU. The training source code and the trained convnets are available at `https://github.com/DagnyT/hardnet`.

## Acknowledgements

The authors were supported by the Czech Science Foundation Project GACR P103/12/G084, the Austrian Ministry for Transport, Innovation and Technology, the Federal Ministry of Science, Research and Economy, and the Province of Upper Austria in the frame of the COMET center, the CTU student grant SGS17/185/OHK3/3T/13, and the MSMT LL1303 ERC-CZ grant. Anastasiya Mishchuk was supported by the Szkocka Research Group Grant.

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
