[Supplementary Material]

# Working Hard to Know Your Neighbors' Margins: Local Descriptor Learning Loss

Anastasiya Mishchuk[1] , Dmytro Mishkin[2] , Filip Radenovic[2], Jiri Matas[2]

1. Szkocka Research Group, Ukraine. anastasiya.mishchuk@gmail.com
2. Visual Recognition Group, Center for Machine Perception, FEE, CTU in Prague. {mishkdmy, filip.radenovic, matas}@cmp.felk.cvut.cz

## Summary

We introduce:
1. **HardNet local feature descriptor** which improves state-oft-the art in wide baseline stereo, patch matching, verification and retrieval and in image retrieval.
2. **HardNet triplet loss for metric learning tasks** which maximizes the distance between the closest positive and closest negative example in the batch. HardNet loss is better than complex regularization methods; it is inspired by SIFT matching scheme.

## HardNet (L2Net Architecture)

## HardNet Triplet Loss

Patches are described by HardNet, then n x n distance matrix D is calculated, n – mini-batch size. For each positive pair (a,p) find the non-matching descriptor closest to either a or p.

Contribution to the gradient magnitude from the positive and negative examples. Horizontal and vertical axes show the distance from the anchor (a) to the negative (n) and positive (p) examples respectively.

$$\frac{\partial L}{\partial n} = \frac{\partial L}{\partial p} = 1$$
$$\frac{\partial L}{\partial n} = 0, \frac{\partial L}{\partial p} = 1$$
$$\frac{\partial L}{\partial n} = \frac{\partial L}{\partial p} = 0$$

## Datasets

**Brown PhotoTour dataset:**
3 sets, 400k DoG patches each:
- Liberty (shown)
- Notre-Dame
- Yosemite
Size: 64x64, grayscale, obtained from SfM model

**HPatches dataset:**
reference image + 5 target images:
V: 57 images – photometric changes
I: 59 images – geometric changes
**WBS and HPatches experiments: descriptors were trained on the Brown Liberty**
**Retrieval: HardNet++ model was trained on all sets of Brown + HPatches**

## Wide baseline stereo examples

### Correct inliers with HardNet

### Correct inliers with RootSIFT

## Wide baseline stereo matching on W1BS

A – appearance, G – geometry, L – illumination, S – sensor
Note: training data is G+L type only

## HPatches Matching and Retrieval

**None of the descriptors is trained on HPatches.** Marker color indicates the level of geometrical noise in: EASY, HARD and TOUGH. Marker type indicates the experimental setup. VIEWPT and ILLUM indicate the type of sequences for matching. + denotes training with data augmentation

## PhotoTour Patch Verification

False positive rate at true positive rate equal to 95% (FPR95), is reported. Lower is better

| Training | Notredame | Yosemite | Liberty | Yosemite | Liberty | Notredame | Mean | |
|---|---|---|---|---|---|---|---|---|
| Test | Liberty | | Notredame | | Yosemite | | FDR | FPR |
| SIFT [9] | 29.84 | | 22.53 | | 27.29 | | | 26.55 |
| MatchNet*[14] | 7.04 | 11.47 | 3.82 | 5.65 | 11.6 | 8.7 | 7.74 | 8.05 |
| TFeat-M* [23] | 7.39 | 10.31 | 3.06 | 3.8 | 8.06 | 7.24 | 6.47 | 6.64 |
| L2Net [24] | 3.64 | 5.29 | 1.15 | 1.62 | 4.43 | 3.30 | | 3.24 |
| HardNet (ours) | **3.06** | **4.27** | **0.96** | **1.4** | **3.04** | **2.53** | **3.00** | **2.54** |
| Augmentation: flip, 90° random rotation | | | | | | | | |
| GLoss+[30] | 3.69 | 4.91 | 0.77 | 1.14 | 3.09 | 2.67 | | 2.71 |
| DC2ch2st+[15] | 4.85 | 7.2 | 1.9 | 2.11 | 5.00 | 4.10 | | 4.19 |
| L2Net+ [24] + | 2.36 | 4.7 | 0.72 | 1.29 | 2.57 | **1.71** | | 2.23 |
| HardNet+ (ours) | **2.28** | **3.25** | **0.57** | **0.96** | **2.13** | 2.22 | **1.97** | **1.9** |

## Image Retrieval on Oxford5k, Paris6k

All results are with spatial verification and query expansion. VS: vocabulary size. SA: single assignment. MA: multiple assignments. mAP is reported

| Method | VS | Oxford5k | | Paris6k | |
|---|---|---|---|---|---|
| | | SA | MA | SA | MA |
| SIFT–BoW [36] | 1M | 78.4 | 82.2 | – | – |
| SIFT–BoW-fVocab [46] | 16M | 74.0 | 84.9 | 73.6 | 82.4 |
| RootSIFT–HQE [43] | 65k | 85.3 | 88.0 | 81.3 | 82.8 |
| HardNet++–HQE | 65k | **86.8** | **88.3** | **82.8** | **84.9** |