[Reviews · NeurIPS 2017]

Reviewer 1



This paper proposes a novel triplet loss for learning local patch descriptors for wide-baseline matching, based on maximizing the distance difference between a matching pair and the closest non-matching patch to either of the two patches from the positive pair. In experimental evaluation, the proposed scheme outperforms other very recent descriptor training methods such as [23,24] and state-of-the-art hand-crafted descriptors such as SIFT and RootSIFT. I find the proposed objective function and top-line performance of the method to be promising. However, at present, the paper has two major shortcomings that make it not quite ready for NIPS: - The presentation of the method in Section 3.1 is sloppy and overly confusing. There are typos, e.g., in the definition of distance in line 63 it should probably say a_i p_j instead of a_i p_i. It is not clear why the authors use an asymmetric notation (a and p, i* and j'). The intuition behind lines 65-66 and eq. 1 is not clearly explained. I find Figure 1 to be somewhat helpful but Figure 2 to be completely mysterious. A more geometric illustration of the quadruplets (a_i, p_i, p_{j'}, a_{i*}) and resulting triplets -- or a clearer explanation with a more streamlined notation -- would be better. - There is no proper ablation study comparing the effect of the proposed triplet sampling scheme with other baselines, while keeping all the other elements of the method the same. The experiments are extensive, but they only compare the proposed method with other published systems that may be different in multiple respects. In particular, while L2Net [24] apparently uses the same architecture, its objective function has a completely different form. I would like to see a comparison with the same architecture and basic form of objective function but more "vanilla" pair or triplet terms and sampling schemes. Without such an apples-to-apples comparison, it is impossible to say for sure whether it is really the proposed triplet definition or some other implementation details that account for the good performance of the method. If the authors provide such comparisons in the rebuttal, I may be willing to change my mind to favor acceptance. Post-rebuttal comment: I agree with R3 that the rebuttal is somewhat persuasive and am raising my rating accordingly. I will not object if this paper is accepted to NIPS, although I still can't help feeling that the authors should have explained their contribution more clearly and better distinguished their work from other methods that are on the surface very closely related. It is also unclear to me whether the proposed hard negative mining technique is only effective for the application of learning local descriptors for patch matching, which would make it of fairly limited interest to the NIPS audience, or whether it can be useful more generally. Once again, I agree with R3 in desiring to see major changes to the paper before it can be published (which, of course, is not possible in the NIPS review process).

Reviewer 2



The paper presents a variant of patch descriptor learning using neural networks and a triplet loss. While many similar approaches exist, the particular variant proposed here appears to have better results in a large number of benchmarks. Still, I am really unsure about the technical contribution as the paper is not very clear on this point. The approach appears to be very similar to others such as Balntas 16 that also used triplet losses and deep nets to learn patch descriptors. It seems that the main difference is to consider in the loss hard negative examples. However, Balntas 16 *also* uses hard negatives in the triplets (not merely random samples as argued here on line 46). So what is the difference that changes the empirical results so much? The paper generally needs far more polish. Experiments should carefully describe the difference between different approaches (e.g. descriptor dimensionally, neural network architecture, training set, and anything else that could make a difference). Then, the main reason for the observed empirical boost should be unequivocally identified through empirical assessment. For example, if the claim is that the key is to pool hard negatives, a carefully ablation study comparing this to alternative tripled-formation strategies should be included. Part of such a study may be included in the HardTFeat vs TFeat experiment of Fig. 7, but I would liket to check with the authors that they ran both methods using the same code base and only changing the feature sampling strategy. If more than one thing changes at a time, is difficult to reach a conclusion. If the authors could clearly identify an interesting reason explaining their empirical boost (e.g. they may conclude that the key is a new and better way of doing hard negative mining compared to what has been done so far), then the paper could be interesting enough for acceptance. If, however, the boost is due to other differences such as more tuning or a tweaked architecture, then there would be much less of a reason to accept this into NIPS. Rebuttal: the authors provided an informative rebuttal. One of my key question was about the difference with Blantas 16. The authors argue that Blantas 16 does *not* do hard negative mining. On re-reading Blantas 16, this is partially true: their point is that a simplified form of hard negative mining (called in-triplet hard negatives) is just as effective and in fact superior to hard negative mining, tested in e.g. "Discriminative learning of deep convolutional feature point descriptors". Given the new experiments in the rebuttal, it seems that the main idea here is a new variant of negative mining, within each batch rather than in the whole datasets as done in "Discriminative learning of deep convolutional feature point descriptors", which seems to be similar to the HNM method described in the rebuttal. Hence to me the main message of the paper is that, while HB and HNM are very simple and very similar approaches, HB is in practice far better than HNM. The authors should modify the paper to include a careful experimental analysis of this point, extending and consolidating the new experiments in the rebuttal. With this and other promised improvements, the paper would be good enough for acceptance in my opinion. However, the modifications required from the submitted version are fairly large.

Reviewer 3



This paper proposes an approach to learn local feature descriptors with a CNN. In other word, the point is to defeat the seemingly invincible SIFT [19]. In particular, the goal is to learn a descriptor that is invariant enough while still being discriminative for accurate pairwise matching of interest points. The novelty in this paper lies in the usage of a novel loss, whose goal is to maximize the distance between the closest positive and closest negative. The principle is similar to the "first-to-second nearest distance ratio" rule used in SIFT to determine good descriptor matches from bad matches. I find the paper really good. In a nutshell: the paper is well written; the problem is well defined; the idea is novel, simple and effective; experimental comparisons are plentiful, convincing and done on actual CV tasks (image retrieval, wide baseline stereo) in addition to the usual intermediary evaluation on patch matching/descriptor retrieval. The quality of experiments is at the level of a CVPR paper. There are comparisons with many recent state-of-the-art approaches, on recent datasets, on most matching-based tasks. Results show the consistent superiority of the proposed approach compared to existing methods. What's more, the approach seems easy to replicate and doesn't require additional data for training. Some minor problems: a bit of opacity in Section 3.1: - In eq (1), min(dij,dji) is not clearly defined. I understand that it corresponds to min(d(ai,pj'),d(pi,ai*)) mentioned just above, but the change of notations is confusing. - speaking of which, why use a different superscript (prime and star) for i* and j'? They correspond to similar quantities and are defined almost the same way. - it is not clear what Figure 2 is showing. After spending some time, I can finally understand, but it would benefit the paper it was better explained.